# Origin identification of *Cornus officinalis* based on PCA-SVM combined model

**Yueqiang Jin**[1]*, **Bing Liu**[1], **Chaoning Li**[2], **Shasha Shi**[3]

**1** Public Foundational Courses Department, Nanjing Vocational University of Industry Technology, Nanjing, China, **2** Research and Development Department, Nanjing Changxingyang Intelligent Home Company Limited, Nanjing, China, **3** School of Science, Jiangsu Ocean University, Lianyungang, China

* 3049456656@qq.com

**Data Availability Statement:** All relevant data are within the paper and its Supporting Information files.

## Abstract

Infrared spectroscopy can quickly and non-destructively extract analytical information from samples. It can be applied to the authenticity identification of various Chinese herbal medicines, the prediction of the mixing amount of defective products, and the analysis of the origin. In this paper, the spectral information of *Cornus officinalis* from 11 origins was used as the research object, and the origin identification model of *Cornus officinalis* based on mid-infrared spectroscopy was established. First, principal component analysis was used to extract the absorbance data of *Cornus officinalis* in the wavenumber range of 551~3998 cm$^{-1}$. The extracted principal components contain more than 99.8% of the information of the original data. Second, the extracted principal component information was used as input, and the origin category was used as output, and the origin identification model was trained with the help of support vector machine. In this paper, this combined model is called PCA-SVM combined model. Finally, the generalization ability of the PCA-SVM model is evaluated through an external test set. The three indicators of Accuracy, F1-Score, and Kappa coefficient are used to compare this model with other commonly used classification models such as naive Bayes model, decision trees, linear discriminant analysis, radial basis function neural network and partial least square discriminant analysis. The results show that PCA-SVM model is superior to other commonly used models in accuracy, F1 score and Kappa coefficient. In addition, compared with the SVM model with full spectrum data, the PCA-SVM model not only reduces the redundant variables in the model, but also has higher accuracy. Using this model to identify the origin of *Cornus officinalis*, the accuracy rate is 84.8%.

## 1. Introduction

China has superior climatic and geographical conditions for cultivating herbal medicines. Many of the herbal medicines it cultivates are well known at home and abroad and are exported to more than 100 countries and regions [1, 2]. Compared with synthetic drugs, Chinese herbal medicines have many unique advantages such as natural raw materials, stable effects and less toxic side effects [3]. It is because of these unique advantages that Chinese herbal medicine is receiving more and more attention from countries and regions [4–6].

**Funding:** This work was supported by the research project on philosophy and social science of universities in Jiangsu Province under Grant number 2022SJYB0562 (to Bing Liu) and the horizontal scientific research project of Nanjing Vocational University of Industry Technology under Grant number HK22-38-01 (to Bing Liu). The funders had no role in study design, data collection and analysis, decision to publish, or preparation of the manuscript.

**Competing interests:** The authors have declared that no competing interests exist.

However, with the gradual deterioration of the natural ecological environment, the growth environment of wild Chinese herbs has been destroyed, and some wild Chinese herbs are in short supply, leading to confusion in the Chinese herbal medicine sales market. In addition, the wide variety of Chinese herbal medicines and the different habits of use in different regions make the confusion of Chinese herbal medicines common and the identification of Chinese herbal medicines becomes difficult [7, 8].

The commonly used methods for the identification of herbal medicines include character identification, microscopic identification and physicochemical identification [9, 10]. Character identification is mainly achieved by external characteristics such as appearance, color and odor of herbs, and if necessary, water and fire tests can also be performed. This identification method is easy to operate and can achieve rapid identification, but it requires extensive experience of workers engaged in Chinese medicine identification, and the accuracy of identification of herbs with close relatives and high biomorphological similarity needs to be improved [11, 12]. Microscopic identification is to observe the microscopic structure of herbal medicines through microscope. Each herb has its own special structure, and the microscopic structure of herbs can be observed through the microscope to identify the authenticity of herbs. This method is more commonly used when the shape of the herb is not easily identified, or when the herb is broken or in powder form. However, microscopic identification also has some shortcomings, such as the microscopic characteristics of some herbs are not easy to search, and the identification characteristics of some herbs do not conform to the pharmacopoeia. Physicochemical identification refers to the method of identifying the authenticity, purity and degree of quality merit of herbal medicines by using certain physical, chemical or instrumental analysis methods. Determination of physical constants, determination of swelling, colorimetric examination, foam index, chemical qualitative analysis, and chemical quantitative analysis are all common means of physical and chemical identification. With the development of chromatographic coupling technology, the use of modern chromatographic techniques for the examination of Chinese herbal medicines has also rapidly spread. Commonly used chromatographic techniques include thin layer chromatography [13, 14], gas chromatography [15, 16], high performance liquid chromatography [17–20] and capillary electrophoresis [21–23]. They can achieve very high accuracy in herb identification, but often require complex pretreatment and long analysis time, high analytical cost, and difficulty in non-destructive and rapid identification [24–27].

Infrared spectroscopy has been widely used in the structural analysis of organic compounds [28, 29]. Under infrared irradiation, the molecules of the substance under test only absorb infrared spectra that are consistent with their molecular vibration and rotation frequencies. Therefore, the infrared spectroscopy can be used for the qualitative analysis of the measured substance. As there are many groups in the compound molecule, each group will produce characteristic vibration after excitation, and its vibration frequency will be reflected in the infrared absorption spectrum. Therefore, it can be quantitatively analyzed according to the absorption vibration frequency of various groups in the compound. Chinese herbal medicine is a mixture system composed of many chemical substances [30]. As long as the chemical components contained in the complex system are the same, and the relative proportion between the components is certain, the infrared spectrum obtained is the superposition of the spectra of all the compounds in the system. This superimposed spectrum can be reproducible in a stable manner, just like the spectrum of a single compound. If there is any change in the composition or content of the sample, there will be obvious differences in the spectrum, which provides an objective and reliable basis for the identification and evaluation of the authenticity of the sample.

Infrared spectroscopy has the advantages of fast analysis, low cost, non-destructive and simple pre-treatment. In recent years, it has been widely used in the field of quality control of herbal medicines [31]. Li, W., et al. proposed a discriminant analysis technique for near-infrared spectral classification using wavelet transform and influence matrix analysis methods [32]. This discriminant analysis technique was found to achieve good classification results by testing on the near infrared spectroscopy dataset of 265 salviae miltiorrhizae radix samples from 9 different geographical origins. Lu, L., et al. used Fourier transform infrared spectra combined with pattern recognition technology for geographic identification of wild Gentiana rigescens [33]. The comparison result showed that the Partial Least Squares Discriminant Analysis (PLS-DA) method is more suitable for geographic origin classification of wild Gentiana rigescens than Principal Component Analysis (PCA).

*Cornus officinalis* is a commonly used herbal medicine mainly distributed in central and southern Europe, East Asia and eastern North America [34]. The dried and ripe flesh of *Cornus officinalis* has the ability to nourish the liver and kidneys and quench thirst with internal heat. It is a traditional medicine commonly used in Chinese medicine to treat diabetes. In this study, infrared spectral data of *Cornus officinalis* from a total of 11 origins in Shanxi, Jiangsu, Zhejiang, Anhui, Jiangxi, Shandong, Henan, Hunan, Sichuan, Shaanxi and Gansu (OP 1~OP 11) in China were collected. With the infrared spectral data of these samples, principal component analysis and support vector machine were used to develop a model for identifying the origin of *Cornus officinalis*. This origin identification model is called PCA-SVM combined model in this paper. Compared with other commonly used methods such as the naive Bayesian model, decision tree, LDA, Radial Basis Function (RBF) neural network and PLS-DA, PCA-SVM performs well on some common evaluation indicators [35, 36]. The model can not only provide a convenient and accurate method for the rapid identification of the origin of *Cornus officinalis*, but also has some reference significance for the identification of other herbal medicines.

## 2. Material and methods

### 2.1. Data source and preprocessing

Mid-infrared spectroscopy combined with chemometrics can be used for origin identification of Chinese Herbal Medicines. A set of *Cornus officinalis* spectral data measured by Chengdu University of Traditional Chinese Medicine is used to establish a classification and identification model of *Cornus officinalis*. In this experiment, samples of ripe fruit pulp from 11 origins of *Cornus officinalis* were dried at 40°C to constant weight, crushed by a micro plant pulverizer, and passed through a 200-mesh sieve for use. A mass of 2 mg of *Cornus officinalis* sample powder was uniformly mixed with dried KBr crystals at a mass ratio of about 1:60 in an agate mortar for grinding. The thoroughly grinded mixture was compressed into flakes with a tablet machine, and immediately placed in a Nicolet iS50 FT-IR Spectrometer for measurement. The scanning range was 4000~400 cm$^{-1}$, the spectral resolution was 4 cm$^{-1}$, and the number of scans was 64 times. The interference of water and carbon dioxide was excluded before the background and sample spectra were collected. The prepared sample slices were placed in the spectrometer for measurement and data collection. Due to the environment and operation of the experimental instrument, the original spectrum began to have noise, and finally the spectral data between 3998 ~ 551 cm$^{-1}$ was retained.

The reproducibility of the test is evaluated by taking 5 consecutive measurements of a given sample and calculating the Relative Standard Deviation (RSD) of their maximum common peak wave number. Eq (1) is the expression of RSD, where $S$ is the standard deviation and $\bar{x}$ is the corresponding mean value. The RSD of the reproducibility test is determined to be less

than 0.2%, indicating good reproducibility of the test. The repeatability of the test is evaluated by measuring the same sample once by 5 different experimenters, and calculating the RSD of the maximum common peak wave number of 5 measurements. The RSD of the repeatability test is determined to be less than 3%, indicating good repeatability of the test.

$$RSD = \frac{S}{\bar{x}} \times 100\% \tag{1}$$

Data preprocessing is the first step in data modeling. We consider data that are less than 1/3 times the arithmetic mean of the nearest neighboring values on the left and right or greater than 3 times the arithmetic mean of the nearest neighboring values on the left and right as outliers. Outliers and missing values are interpolated by means of mean interpolation in this paper. After dealing with missing values and outliers, the absorbance data of 3448 corresponding bands under spectral illumination were analyzed and summarized. The range of absorbance after summary is -0.00675~1.48696 AU. There are some negative values of absorbance in the last 184 bands, and there are 626 groups of data with negative values in total. This is because the absorbance used in the data in this paper is the value corrected by the instrument. Since the absolute values of these negative values are small, and the total amount of negative values is less than 0.001% of the total amount of data, we keep them. The absorbance of some data is greater than 1 AU, but the maximum value of absorbance does not exceed 1.5 AU. We think that they do not deviate from the Lambert-Beer law, so no special treatment is required.

## 2.2. Principle of origin identification model

Principal component analysis was first introduced by Karl Pearson for non-random variables, and then Harold Hotelling extended this method to the case of random vectors [37]. Principal component analysis is performed with minimal loss of data information. It uses the method of mathematical transformation to convert the given multiple index factors into a few principal components, and then replaces the original multi-dimensional related variables with a few principal component factors [38]. Suppose the research on a certain problem involves $p$ indicators, which are represented by $X_1, X_2, \cdots, X_p$ respectively, and the $p$-dimensional random vector composed of these p indicators is $X = (X_1, X_2, \cdots, X_p)'$. Let the mean of the random vector $X$ be $\mu$ and the covariance matrix be $\Sigma$. Linear transformation of $X$ can form a new comprehensive variable, which is represented by $Y$, that is, the new variable can be linearly represented by the original variable (Eq (2)).

$$\begin{cases} Y_1 = u_{11}X_1 + u_{21}X_2 + \cdots + u_{p1}X_p \\ Y_2 = u_{12}X_1 + u_{22}X_2 + \cdots + u_{p2}X_p \\ \quad\quad\quad\quad\quad\vdots \\ Y_p = u_{1p}X_1 + u_{2p}X_2 + \cdots + u_{pp}X_p \end{cases} \tag{2}$$

The above linear transformation of the original variable can be carried out arbitrarily, and the statistical characteristics of the comprehensive variable Y obtained by different linear transformations are also different. Therefore, in order to achieve better results, we hope that the variance of $Y_i = u_i'X$ is as large as possible and each $Y_i$ is independent of each other. Since $var(Y_i) = var(u_i'X) = u_i'\sum u_i$, and for any constant $c$, there is $var(cu_i'X) = c^2u_i'\sum u_i$, so when there is no restriction on $u_i$, $var(Y_i)$ can be increased arbitrarily, and the problem will become meaningless. We constrain linear transformations to the following principles: (i) $u_i'u_i = 1$ ($i = 1, 2, \cdots, p$); (ii) $Y_i$ and $Y_j$ are independent of each other ($i \neq j$; $i, j = 1, 2, \cdots, p$); (iii) $Y_1$ is the one with the largest variance among all linear combinations of $X_1, X_2, \cdots, X_p$ that satisfy

the principle (i); $Y_2$ is the one with the largest variance among all the linear combinations of $X_1, X_2, \cdots, X_p$ that are not related to $Y_1$; $\cdots$; and $Y_p$ is the one with the largest variance among all linear combinations of $X_1, X_2, \cdots, X_p$ that are not related to $Y_1, Y_2, \cdots, Y_{p-1}$. The comprehensive variables $Y_1, Y_2, \cdots, Y_p$ determined based on the above three principles are called the first, second, $\cdots$, $p$-th principal components of the original variables. In actual research work, only the first few principal components with the largest variance are usually selected, so as to simplify the system structure and grasp the essence of the problem.

$$\max_{w,b} \frac{2}{\parallel w \parallel}$$

$$s.t. \ y_i(w^T x + b) \geq 1, \quad i = 1, 2, \cdots, m \tag{3}$$

$$\min_{w,b} \frac{1}{2} \parallel w \parallel^2$$

$$s.t. \ 1 - y_i(w^T x + b) \leq 0, \quad i = 1, 2, \cdots, m \tag{4}$$

The support vector machine proposed by Cortes and Vapnik is an algorithm to find a classification plane or hyperplane that separates different types of data in the dataset as much as possible [39]. Fig 1 is its architecture. SVM has the theory of structural risk minimization, and it still has strong robustness in the face of nonlinear datasets and higher-dimensional datasets, so

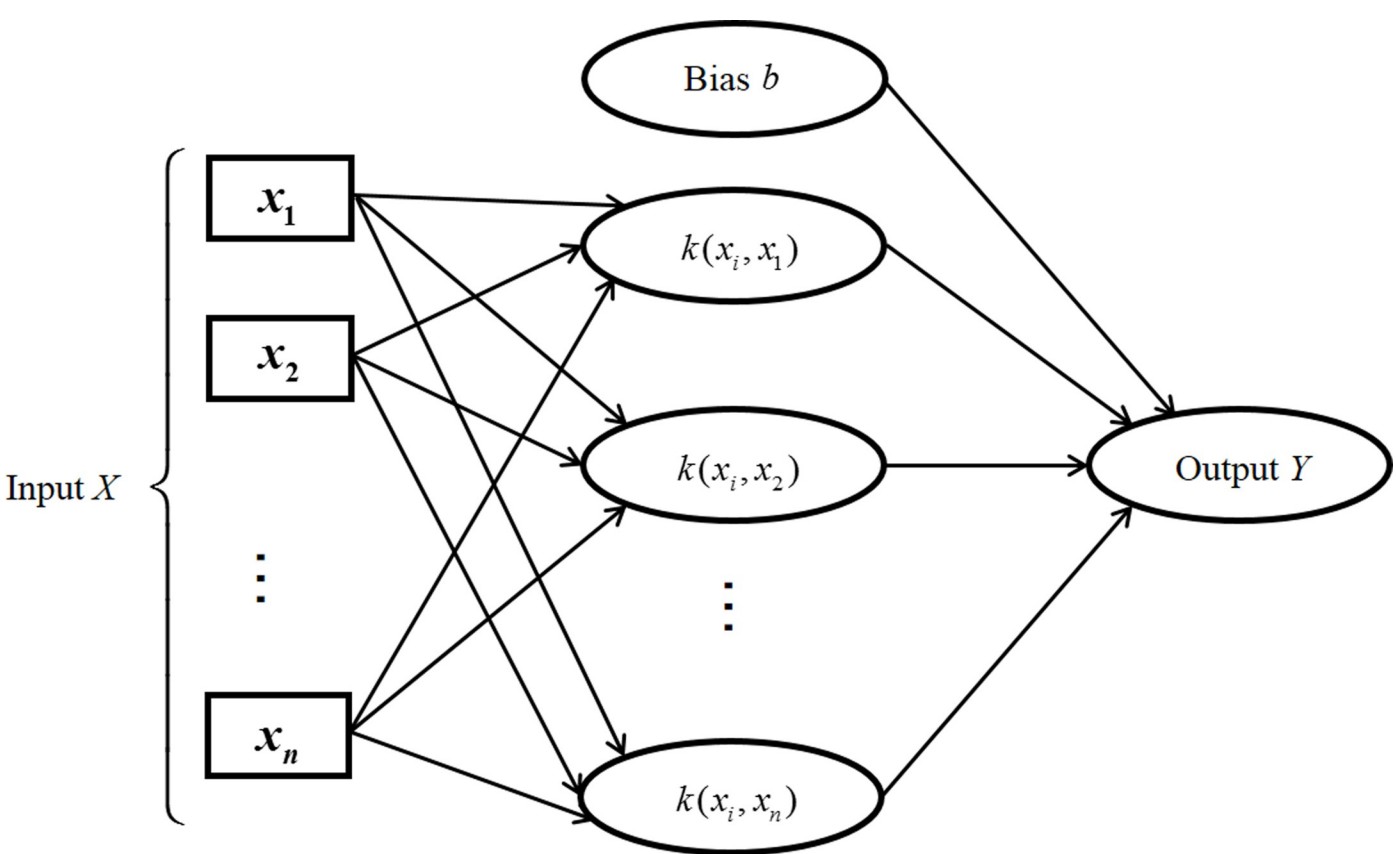

**Fig 1. Support vector machine architecture.**

it is widely used in classification algorithms [40]. The essence of SVM is to obtain the optimal parameters $w$ and $b$ to determine an optimal hyperplane, so that as much data as possible is distributed on both sides of this plane to achieve classification. Assuming that the training set is $(x_i, y_i)$, $i = 1, 2, \cdots, l$, $x \in R^n$, $y \in \{\pm 1\}$, the linear equation $w^T x + b = 0$ is used to divide it, where $w = (w_1; w_2; \cdots; w_d)$ is the normal vector and $b$ is the bias term. In order for the hyperplane to have maximum margin (Eq (3)), it is only necessary to maximize $\|w\|^{-1}$, which is equivalent to minimizing $\|w\|^2$. Therefore, the problem of constructing the optimal hyperplane is transformed into Eq (4).

Eqs (5)–(6) are obtained by introducing Lagrange multipliers, where $\alpha = (\alpha_1; \alpha_2; \cdots; \alpha_m)$. Eqs (7)–(8) can be obtained by taking the partial derivatives of $w$ and $b$ of $L(w, b, \alpha)$. Substituting Eq (6) into Eq (5), $w$ and $b$ in $L(w, b, \alpha)$ can be eliminated, and then considering the constraints in Eq (7), the dual problem of Eq (3) can be obtained (Eq (9)).

$$min \; maxL(w, b, \alpha) \tag{5}$$

$$L(w, b, \alpha) = \frac{1}{2}\| w \|^2 + \sum_{i=1}^{m} \alpha_i (1 - y_i(w^T x + b)) \tag{6}$$

$$w = \sum_{i=1}^{m} \alpha_i y_i x_i \tag{7}$$

$$0 = \sum_{i=1}^{m} \alpha_i y_i \tag{8}$$

$$\max_{\alpha} \sum_{i=1}^{m} \alpha_i - \frac{1}{2} \sum_{i=1}^{m} \sum_{j=1}^{m} \alpha_i \alpha_j y_i y_j x_i^T x_j$$

$$s.t. \quad \sum_{i=1}^{m} \alpha_i y_i = 0, \quad \alpha_i \geq 0, \quad i = 1, 2, \cdots, m \tag{9}$$

## 3. Establishment of origin identification model

### 3.1. Data exploratory analysis

Exploratory analysis can give us a preliminary understanding of the spectral data of the sample. Fig 2 shows the mid-infrared spectra of 658 samples. It can be seen from the spectrum comparison that the mid-infrared spectrum of *Cornus officinalis* has obvious similarity, especially in the range of 1700~2500 cm$^{-1}$. In the two band ranges of 1000~1700 cm$^{-1}$ and 3000~3400 cm$^{-1}$, there are mainly strong spectral peaks, and the peaks change drastically. This spectral region contains more chemical information. Five strong peaks appeared near the 1070, 1400, 1700, 2950 and 3300 cm$^{-1}$ bands, with average absorbances of 0.734, 0.508, 0.781, 0.455 and 0.827 AU. The spectrum fluctuates greatly in the 1600~1700 cm$^{-1}$ and 3250~3350 cm$^{-1}$ bands, indicating that there are certain differences in the mid-infrared spectra of *Cornus officinalis* from different origins. The difference in absorbance of different origins can be used to identify the origin of *Cornus officinalis* [19, 22].

In order to further compare the differences in mid-infrared spectra from different origins, we classified and summarized 658 samples by category, and averaged the absorbance under different wavelength bands. Fig 3 shows the average absorbance of *Cornus officinalis* from 11 origins in different wavelength bands. It can be seen that the spectral averages of *Cornus*

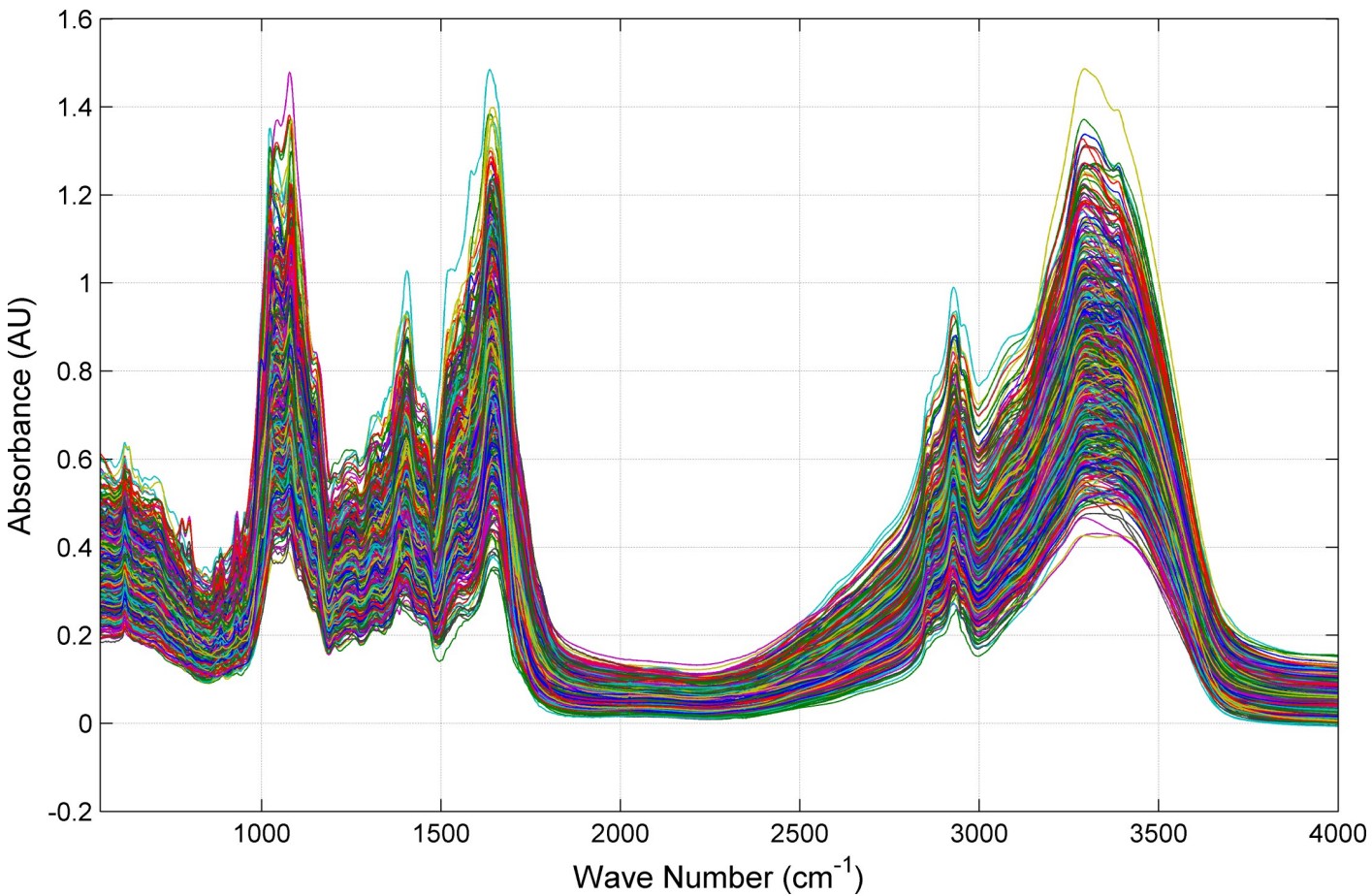

**Fig 2. Mid-infrared spectral of 658** *Cornus officinalis* **samples from 11 different places of origin.** Figs are generated using Matlab (Version R2021b, https://www.mathworks.com/) [Software].

*officinalis* from the 11 origins are very similar, and the bands where the spectral peaks appear are basically the same. The C−O stretching vibration absorption peak is around 1000~1250 cm$^{-1}$, the aromatic ring skeleton vibration absorption peak is around 1400~1600 cm$^{-1}$, the carbonyl C = O stretching vibration absorption peak is around 1700 cm$^{-1}$, the methylene C−H antisymmetric stretching vibration absorption peak is around 2950 cm$^{-1}$, and the O−H stretching vibration absorption peak is around 3300 cm$^{-1}$. In some wavelength bands, the absorbance of different origins is different, which is due to the different climate and geographical conditions of different origins.

## 3.2. Mid-infrared spectral feature extraction

Mid-infrared spectroscopy was used in this study to identify the origin of *Cornus officinalis*. Our collection of mid-infrared spectral data contains 3448 bands, and the absorbance data for each band are highly correlated. If all 3448 variables are introduced into the *Cornus officinalis* origin identification model, it will not only make the model training time very long, but also the introduction of highly correlated variables into the model will lead to poor stability and generalization ability. Therefore, it is necessary to use mathematical methods to extract features from the data. Principal component analysis is a common unsupervised analysis technique that is often used to extract features from complex data [32, 41].

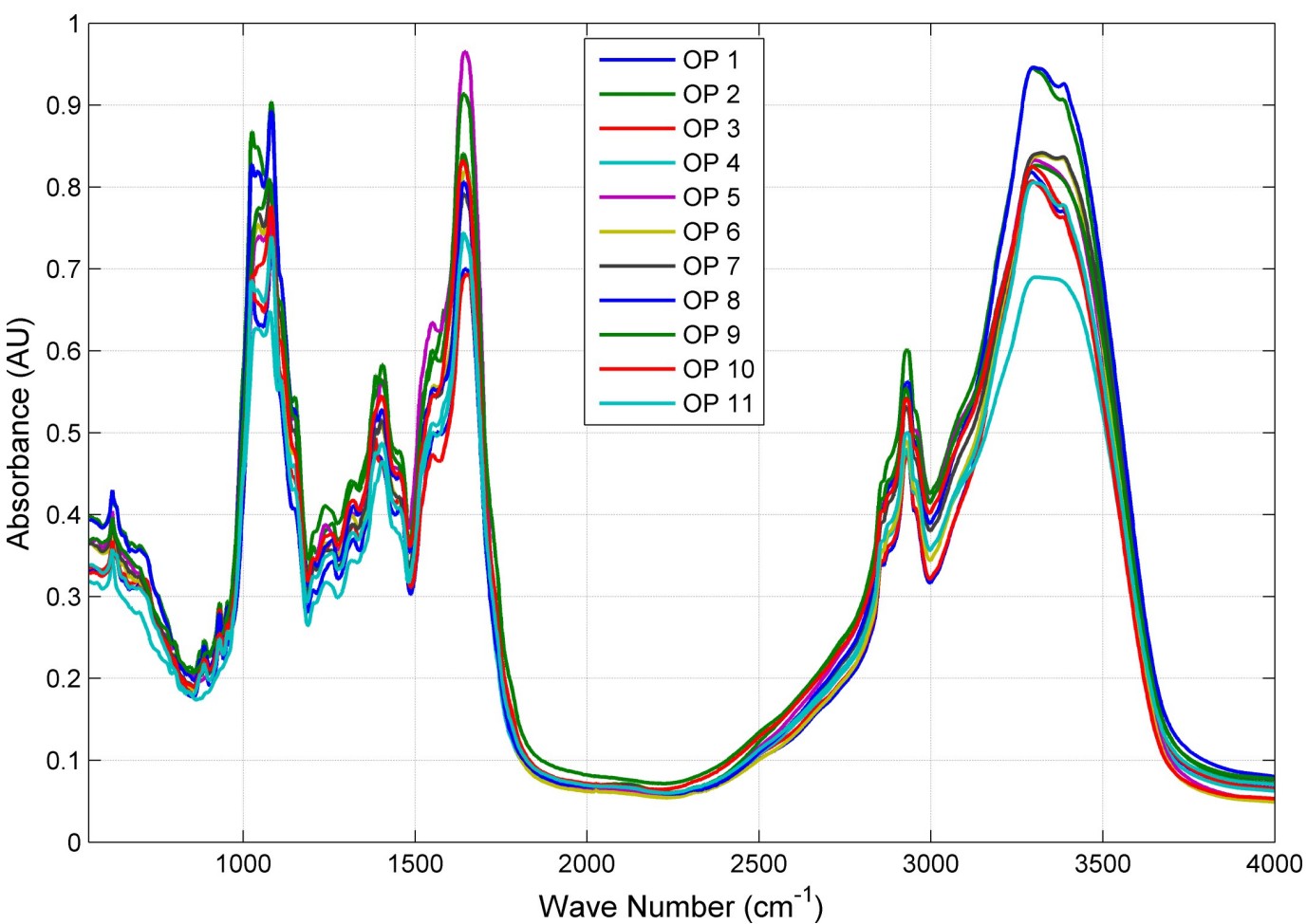

**Fig 3. The average mid-infrared spectra of *Cornus officinalis* samples by different places of origin.**

Table 1 shows the results of principal component analysis of the mid-infrared spectral data of *Cornus officinalis*. It can be seen that the first principal component contains 80.8% of the information of the original data, and the first three principal components contain more than 95% of the information of the original data. Common methods for the selection of the number of principal components include the cumulative contribution rate criterion and the Kaiser criterion based on eigenvalues greater than 1 [42]. In order to introduce more variables into the model so that the model can be fully trained, this paper adopts the Kaiser criterion to select the number of principal components. The eigenvalues corresponding to the first 14 principal components are all greater than 1, and their cumulative contribution rate exceeds 99.8%, capturing most of the information of the original variables. According to the Kaiser criterion, the first 14 principal components are selected to replace the original spectral data to establish the origin identification model of *Cornus officinalis*.

### 3.3. PCA-SVM combined model construction

The 14 principal components after feature extraction are used to establish the origin identification model of *Cornus officinalis*. Since SVM can usually get better results than other algorithms such as naive Bayes, decision trees and linear discriminant analysis on a small sample training

**Table 1. Principal component eigenvalues, contribution rate and cumulative contribution rate of the mid-infrared spectral data of *Cornus officinalis*.**

| Serial number | Principal component | Eigenvalues | Contribution rate (%) | Cumulative contribution rate (%) |
|---|---|---|---|---|
| 1 | 1st principal component | 2785 | 0.808 | 0.808 |
| 2 | 2nd principal component | 314.31 | 0.091 | 0.899 |
| 3 | 3rd principal component | 174.91 | 0.051 | 0.950 |
| 4 | 4th principal component | 77.257 | 0.022 | 0.972 |
| 5 | 5th principal component | 29.467 | 0.009 | 0.981 |
| 6 | 6th principal component | 18.32 | 0.005 | 0.986 |
| 7 | 7th principal component | 15.433 | 0.004 | 0.990 |
| 8 | 8th principal component | 9.94 | 0.003 | 0.993 |
| 9 | 9th principal component | 7.689 | 0.002 | 0.995 |
| 10 | 10th principal component | 2.614 | 0.001 | 0.996 |
| 11 | 11th principal component | 2.174 | 0.001 | 0.997 |
| 12 | 12th principal component | 1.761 | 0.001 | 0.997 |
| 13 | 13th principal component | 1.446 | 0.000 | 0.998 |
| 14 | 14th principal component | 1.113 | 0.000 | 0.998 |
| 15 | 15th principal component | 0.928 | 0.000 | 0.998 |

set and has better robustness, it is used to establish a mid-infrared spectroscopy-based identification model for the origin of *Cornus officinalis* [15, 43, 44].

The 658 samples are sorted by the Matlab random permutation function and divided into two parts in a ratio of approximately 3:1. The first part contains 500 samples for training the model and the second part contains 158 samples for testing the model. Since the sample data comes from 11 different origins, and the number of samples in each origin is different, it is necessary to test the balance of the samples. Fig 4 shows the number of samples included in the training set, test set, and all sets under each origin category. It can be seen that the number of OP 4 in all sets is at most 88, accounting for 13.4% of the total number of samples, and the number of OP 5 is at least 31, accounting for 4.7% of the total number of samples. The number of OP 6 in training set is at most 72, accounting for 14.4% of the total number of samples, and the number of OP 5 is at least 20, accounting for 4% of the total number of samples. The number of OP 4 in the test set is at most 24, accounting for 15.2% of the total number of samples, and the number of OP 9 is at least 7, accounting for 4.4% of the total number of samples. No matter which set, the ratio of the maximum number and the minimum number of samples does not exceed 4:1, so it is considered that there is no sample imbalance in this study.

A model between the origin identification of *Cornus officinalis* and its discriminant index was established, and the training samples are grouped by K-fold Cross-Validation (KCV) method. K-fold cross-validation is a statistical analysis method used to verify the performance of a classifier [45]. Its basic idea is to group the original data, one part as the training set and the other part as the validation set. First train the classifier with the training set, and then use the validation set to test the trained model, which is used as the performance indicator for evaluating the classifier. KCV divides the original data into K groups, extracts a subset without repetition as a validation set, and combines the remaining K-1 sets of subset data as a training set, as shown in Fig 5. In this paper, the 10-fold cross-validation method is selected.

The samples are grouped by the K-fold cross-validation method, and then cross-trained by SVM to construct the SVM-based identification model of *Cornus officinalis*. Its specific process is shown in Fig 6. (i) Normalization of extracted principal components. (ii) The sample data is grouped for training by the K-fold cross-validation method, and K = 10 is selected. (iii) Take each subset (50 samples) data as a validation set, and the remaining 9 sets of subset (450

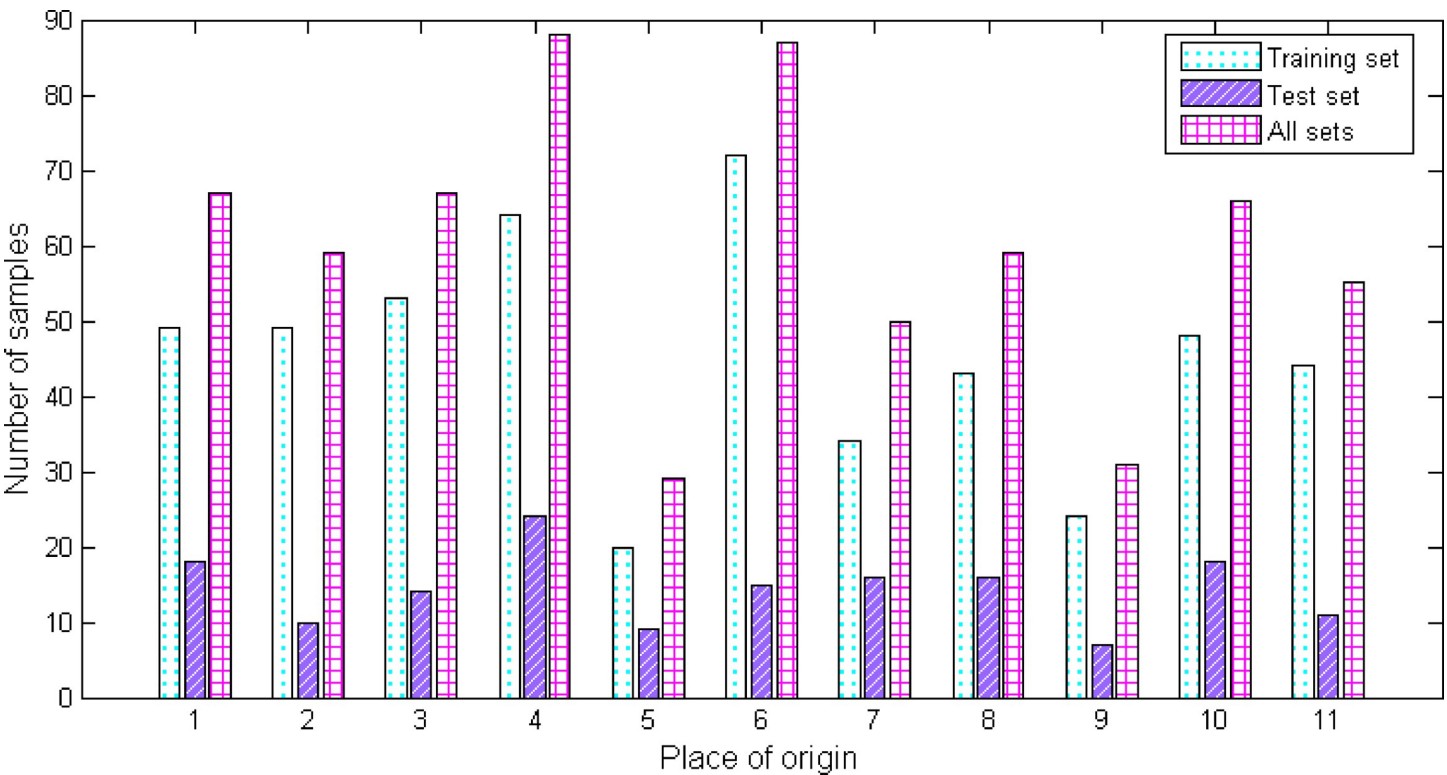

**Fig 4. The number of samples in the training set, test set and all sets of *Cornus officinalis* from different origins.**

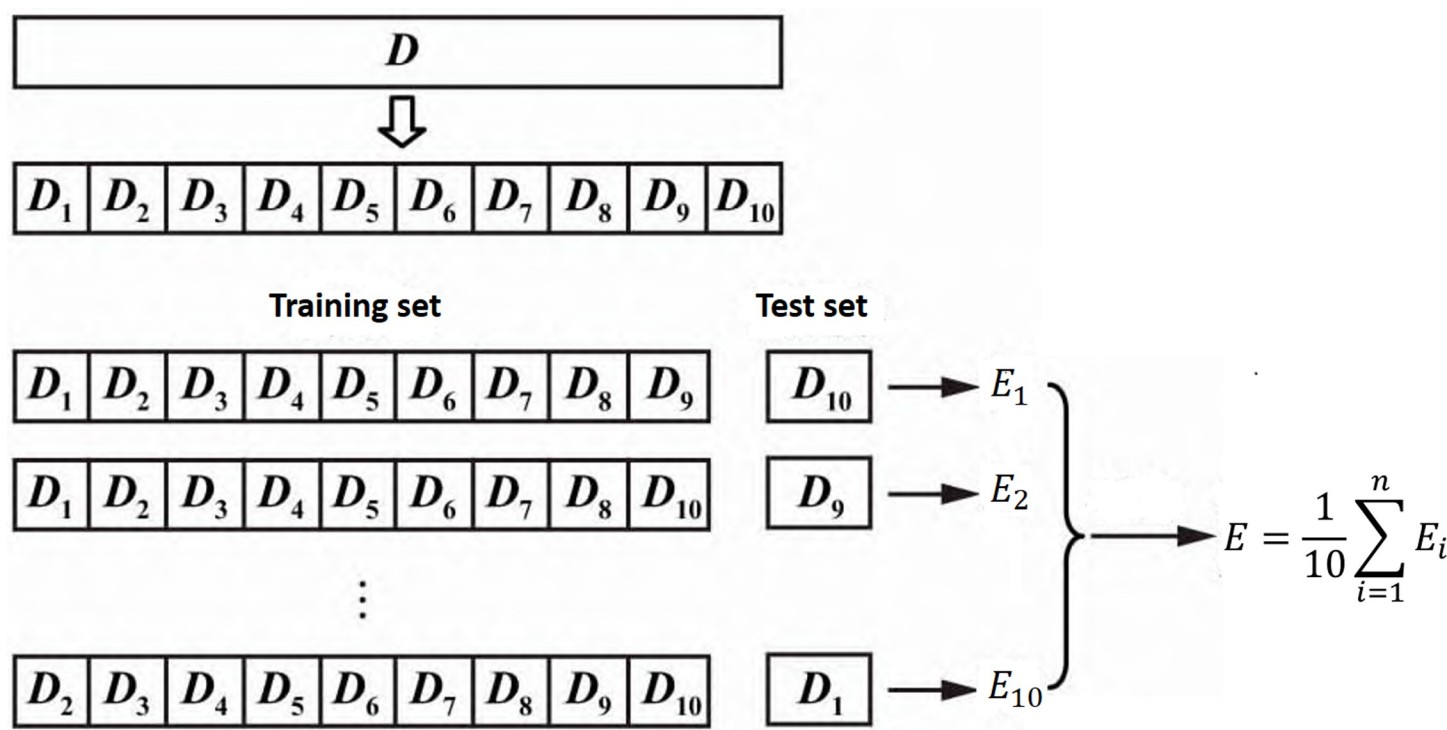

**Fig 5. 10-fold cross-validation process description and implementation.**

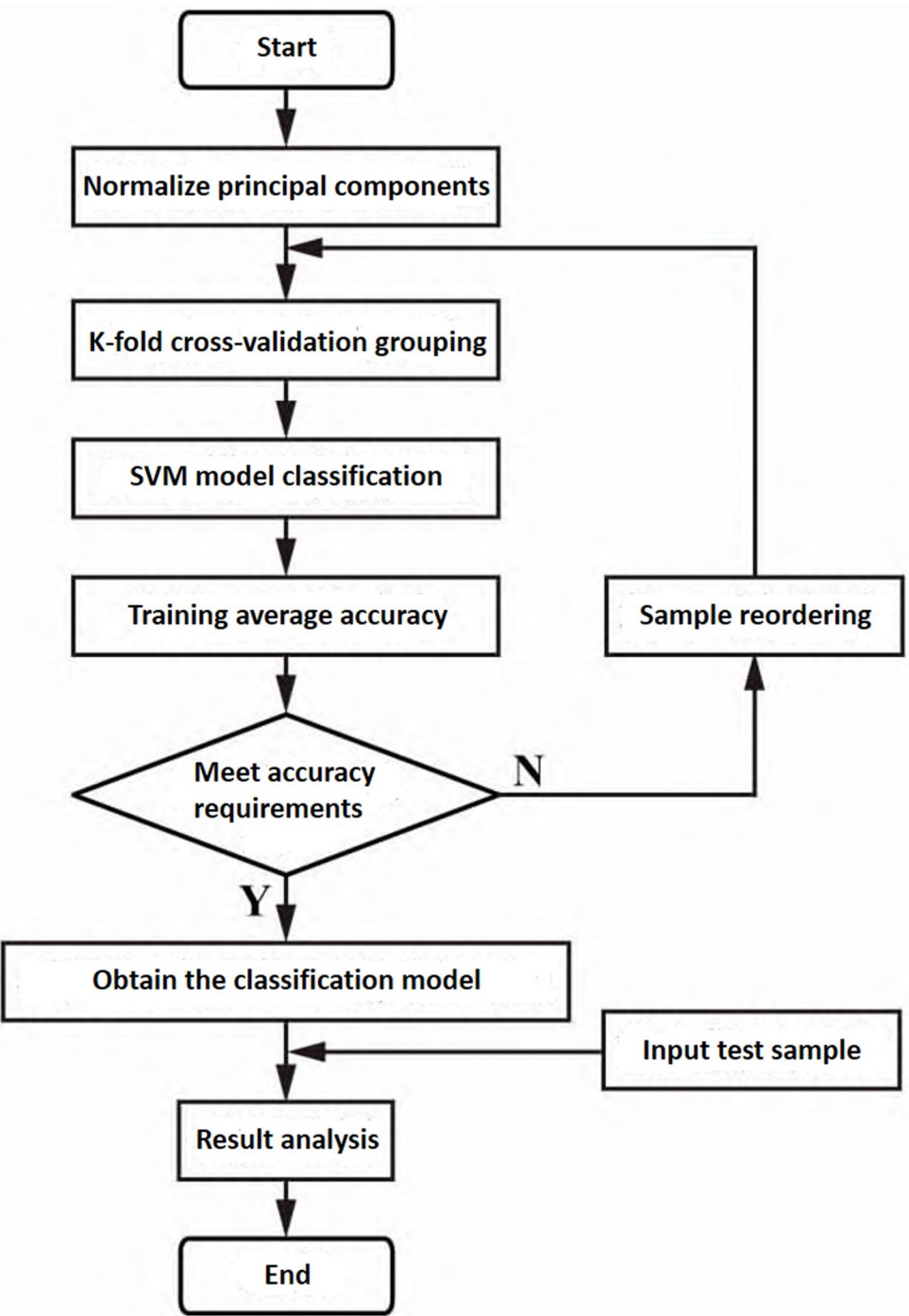

**Fig 6. Flow chart of realization of SVM origin identification model of *Cornus officinalis* based on K-fold cross-validation.**

samples) data as a training set, so that 10 training model data will be obtained and brought into the SVM model for training. (iv) When the average accuracy rate of the model is greater than or equal to 80%, it is determined that the model can identify the origin of *Cornus officinalis*, and the optimal result of the training model is determined as the classification model. If the

average accuracy rate is less than 80%, the samples will be re-sorted randomly, and return to step (ii) to perform K-fold cross-validation. (v) A K-fold cross-validation-based SVM identification model for the origin of *Cornus officinalis* is obtained. (vi) Input the test sample and get the classification result.

The principal components extracted from the spectral data of *Cornus officinalis* are used as input variables, OP is used as output variables, and 500 divided samples are trained with the help of SVM based on k-fold cross-validation. For SVM based on kernel function, this study compares linear kernel, quadratic kernel, cubic kernel and Gaussian kernel. Eqs (10)–(12) are their expressions, where $x$ is the vector drawn from the input space, $x_i$ is the support vector, $\gamma$ the coefficients of the kernel function, $r$ is the constant term in the kernel function, and $p$ is the degree of polynomial kernel functions. The box constraint selects the default value of 1. The smaller the box constraint, the larger the margin, which means that the more error samples allowed in training, the more support vectors, and the stronger the generalization ability. Kernel scale mode select auto, set to auto to use a heuristic procedure to select the scale value using subsampling. For the selection of multiclass method, since there is no sample imbalance in this study, we choose the one-vs-all with higher efficiency. The accuracy on the validation set is used to compare different kernel functions. The accuracy rate of the support vector machine with the quadratic kernel function after the experiment is 82.4% on the validation set, which is the highest among the given kernel functions and meets the accuracy requirements set in advance. With the help of Matlab classification learner (Version R2021b, https://www.mathworks.com/), the PCA-SVM combined model for identifying the origin of *Cornus officinalis* based on mid-infrared spectroscopy is established. For the SVM model with full spectrum data, the same settings were performed, resulting in a maximum accuracy of 57.1% on the validation set, which did not meet the preset accuracy requirement. This is mainly due to the existence of multicollinearity in the full spectrum data, the information input to the model overlaps with each other, and the model is very unstable.

$$K(x, x_i) = x^T x_i \tag{10}$$

$$K(x, x_i) = (\gamma x^T x_i + r)^p, \quad \gamma > 0 \tag{11}$$

$$K(x, x_i) = exp(-\gamma \| x - x_i \|^2), \quad \gamma > 0 \tag{12}$$

In order to verify whether the model has good generalization ability, 158 test sample data onto *Cornus officinalis* of known origin are input into the model for prediction, and the predicted results are compared with the known results. Fig 7 shows the confusion matrix for the test samples. Each row of the confusion matrix represents the predicted category, and the total number of each row represents the number of data predicted for that category; each column represents the true category, and the total number of data in each column represents the number of data instances of that category. The Precision of each category is shown on the left side of the confusion matrix, the Recall (also called Sensitivity) of each category is shown on the lower side of the confusion matrix, and the Accuracy of the model is shown in the lower right corner of the confusion matrix. Eqs (13)–(15) is the equations of Precision, Recall and Accuracy, where TP indicates the number of samples whose true value is positive and the model judges as positive; FN indicates the number of samples whose true value is positive and the model judges as negative; FP indicates the number of samples whose true value is negative and the model judges as positive; TN indicates the number of samples whose true value is negative

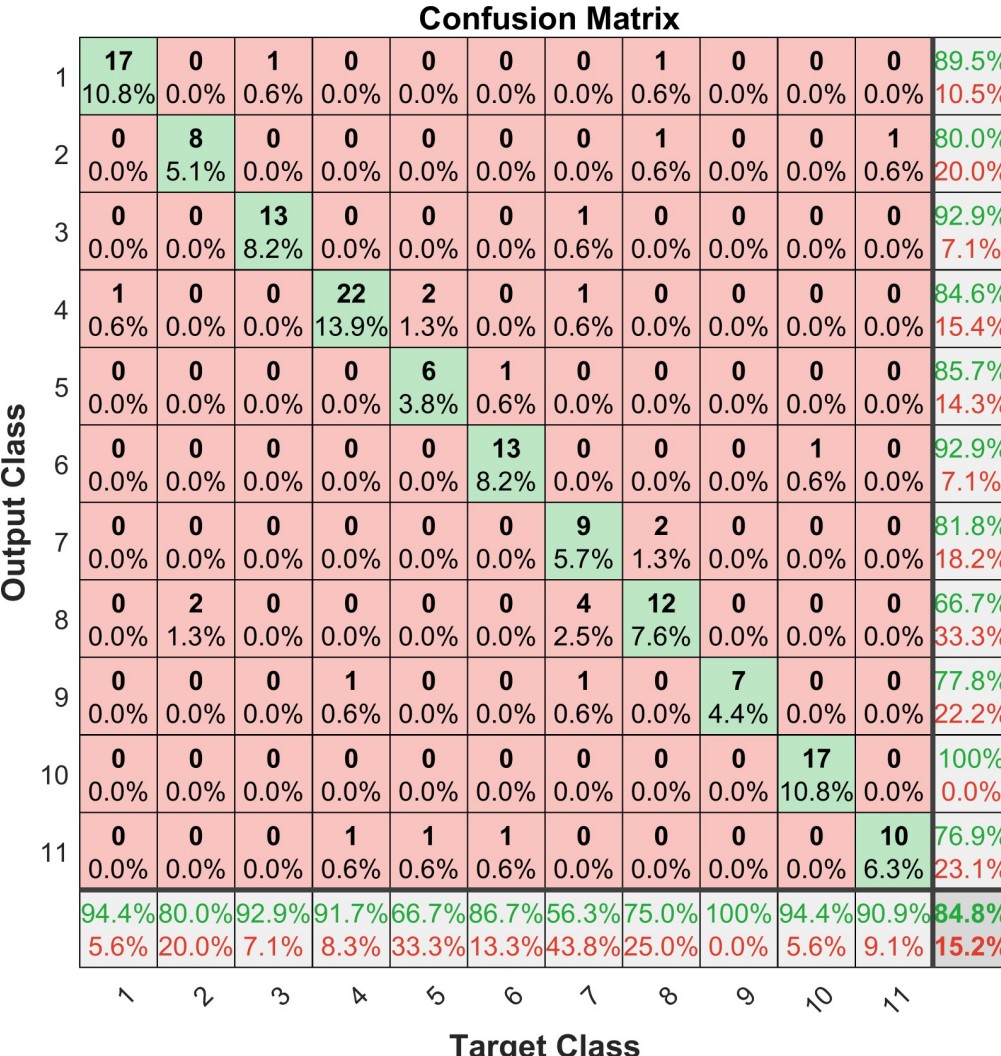

**Fig 7. Confusion matrix for *Cornus officinalis* test samples.** Each row of the confusion matrix represents the predicted category, and each column represents the true category.

and the model judges as negative [40, 46].

$$Precision_i = \frac{TP_i}{TP_i + FP_i} \tag{13}$$

$$Recall_i = \frac{TP_i}{TP_i + FN_i} \tag{14}$$

$$Accuracy = \frac{TP + TN}{TP + TN + FP + FN} \tag{15}$$

It can be seen from the confusion matrix that there are 17 *Cornus officinalis* samples predict by the model as OP10, and they are all from OP10. This category has the highest Precision at 100%. There are 18 *Cornus officinalis* samples predict by the model as OP 8, and 12 of them

are from OP 8. This category has the lowest Precision at 66.7%. Here 4 samples of OP 7 are incorrectly predict to be OP 8 by the model. In terms of Sensitivity, all 7 samples of OP 9 are recognized by the model, and the Recall is the highest at 100%. Only 9 of the 16 samples of OP 7 are identified, and the Recall is the lowest at 56.3%, of which 4 samples are misjudged as OP 8. In general, *Cornus officinalis* from OP 7 and OP 8 are easily confused. This is because the climate and geographical conditions of the two places are similar, resulting in similar chemical composition of *Cornus officinalis*. Among the total 158 samples of *Cornus officinalis*, 134 origins are correctly predicted, and the Accuracy is 84.8%, which is similar to the accuracy rate of the validation set, indicating that the model has a strong generalization ability.

## 4. Discussion

The spectral characteristics of different Chinese Herbal Medicines are quite different. Even the same Chinese Herbal Medicines from different origins will show different spectral characteristics under the irradiation of near-infrared and mid-infrared spectra due to the differences in the chemical composition of inorganic elements and organic matter. Therefore, these characteristics can be used to identify the species and origin of Chinese Herbal Medicines. Based on mid-infrared spectral data, naive Bayes, decision trees, LDA, RBF and PLS-DA can all identify the origin of *Cornus officinalis* [47].

Bayesian methods are based on Bayesian principles and use knowledge of probability statistics to classify sample data sets. The Bayesian approach is characterized by combining prior and posterior probabilities, i.e., it avoids the subjective bias of using only prior probabilities and the overfitting phenomenon of using sample information alone. The naive Bayesian method is a corresponding simplification based on the Bayesian algorithm, that is, it is assumed that the attributes are conditionally independent of each other when the target value is given. Although this simplification reduces the classification effectiveness of Bayesian classification algorithm to some extent, it greatly simplifies the complexity of Bayesian methods in practical application scenarios.

Decision tree is a basic classification and regression method. The decision tree model has a tree-like structure and represents the process of classifying instances based on features in a classification problem. It can be thought of as a set of if-then rules, or as a conditional probability distribution defined in feature space and class space. Its main advantages are the readability of the model and the speed of classification. For learning, a decision tree model is built based on the principle of minimizing the loss function using the training data. For prediction, the decision tree model is used to classify the new data.

Linear discriminant analysis is a classic linear learning method, which was first proposed by Fisher in 1936 on the binary classification problem. The idea of linear discrimination is that for a given set of training samples, we try to project the samples onto a straight line so that the projection points of similar samples are as close as possible and the projection points of dissimilar samples are as far away as possible. When classifying a new sample, it is projected onto the same straight line, and then the class of the new sample is determined based on the location of the projected points.

Broomhead and Lowe first used radial basis functions for neural network design in 1988 [48]. Radial basis function neural network is a commonly used three-layer feedforward network, which can be used for both function approximation and pattern classification. Compared with other types of artificial neural networks, RBF networks have a physiological basis, simple structure, fast learning speed, excellent approximation performance and generalization ability.

Partial least squares regression analysis is a statistical method that is related to principal component regression, but instead of finding the hyperplane of maximum variance between

the response and independent variables, a linear regression model is found by projecting the independent and response variables into a new space, respectively. Because both predictor and response variables are projected into the new space, the methods in the PLS family are called bilinear factorial models [49, 50]. When the response variable is categorical data it is called partial least squares discriminant analysis.

Table 2 shows the Precision and Recall of each model for the identification of *Cornus officinalis* from different origins [43, 51, 52]. It should be noted that the sample division of training set and test set of each model and the method of model validation are the same as those of PCA-SVM combined model. It can be seen that each model has a certain ability to identify the origin of *Cornus officinalis*. Decision tree has the highest precision in OP 4; LDA has the highest recall in OP 5; RBF has the highest precision in OP 8 and OP 11, and the highest recall in OP 11; PLS-DA has the highest precision in OP 5 and OP 9, and the highest recall in OP 8 and OP 10; the PCA-SVM combined model presented in this paper has the highest precision in other origins except OP 4, OP 5, OP 8, OP 9 and OP 11, and the highest recall in other origins except OP 5 and OP 11.

In order to comprehensively compare the ability of each model to identify the origin of *Cornus officinalis*, we compared each model from the three indicators of Accuracy, F1-Score and Kappa coefficient. The F1-Score indicator combines the results of Precision and Recall output. Its value ranges from 0 to 1, where 1 represents the best output of the model, and 0 represents the worst output of the model. F1-Score needs to average each category of Precision and Recall (Eqs (16)–(17)), and then use Eq (18) to calculate.

$$Precision = \frac{\sum_{i=1}^{n} Precision_i}{n} \tag{16}$$

$$Recall = \frac{\sum_{i=1}^{n} Recall_i}{n} \tag{17}$$

$$F1 = 2 \cdot \frac{precision \cdot recall}{precision + recall} \tag{18}$$

$$Kappa = \frac{p_0 - p_e}{1 - p_e} \tag{19}$$

**Table 2. Precision and recall (sensitivity) of each Chinese herbal medicine origin identification model in 11 different origins, where PPV stands for precision and TPR stands for sensitivity (values are measured in %).**

| OP | Naive Bayes | | Decision Trees | | LDA | | RBF | | PLS-DA | | PCA-SVM | |
|---|---|---|---|---|---|---|---|---|---|---|---|---|
| | PPV | TPR | PPV | TPR | PPV | TPR | PPV | TPR | PPV | TPR | PPV | TPR |
| OP 1 | 77.8 | 77.8 | 83.3 | 55.6 | 83.3 | 83.3 | 81.3 | 72.2 | 76.5 | 72.2 | 89.5 | 94.4 |
| OP 2 | 35.7 | 50.0 | 23.5 | 40.0 | 31.8 | 70.0 | 30.0 | 60.0 | 55.6 | 50.0 | 80.0 | 80.0 |
| OP 3 | 80.0 | 85.7 | 61.5 | 57.1 | 85.7 | 85.7 | 73.3 | 78.6 | 90.9 | 71.4 | 92.9 | 92.9 |
| OP 4 | 77.8 | 87.5 | 100 | 75.0 | 91.3 | 87.5 | 86.4 | 79.2 | 77.8 | 87.5 | 84.6 | 91.7 |
| OP 5 | 85.7 | 66.7 | 50.0 | 66.7 | 80.0 | 88.9 | 77.8 | 77.8 | 100 | 66.7 | 85.7 | 66.7 |
| OP 6 | 41.2 | 46.7 | 50.0 | 73.3 | 80.0 | 53.3 | 52.6 | 66.7 | 58.8 | 66.7 | 92.9 | 86.7 |
| OP 7 | 60.0 | 37.5 | 35.7 | 31.3 | 30.0 | 18.8 | 0.0 | 0.0 | 75.0 | 56.2 | 81.8 | 56.3 |
| OP 8 | 76.9 | 62.5 | 69.2 | 56.3 | 76.9 | 62.5 | 92.3 | 66.7 | 60.0 | 75.0 | 66.7 | 75.0 |
| OP 9 | 55.6 | 71.4 | 66.7 | 57.1 | 87.5 | 100 | 44.4 | 72.7 | 100 | 71.4 | 77.8 | 100 |
| OP10 | 92.3 | 66.7 | 61.9 | 72.2 | 82.4 | 77.8 | 52.9 | 56.2 | 73.9 | 94.4 | 100 | 94.4 |
| OP11 | 46.7 | 63.6 | 40.0 | 36.4 | 50.0 | 63.6 | 77.8 | 100 | 72.7 | 72.7 | 76.9 | 90.9 |

The Kappa coefficient is an indicator used for consistency checks and can also be used to measure the effect of classification. Its calculation is based on the confusion matrix, which takes values between -1 and 1, usually greater than 0. Eq (19) is the calculation equation of the Kappa coefficient, where $p_0 = \sum_i p_{ii}$ is called the observation concordance rate, $p_{ii} = \frac{a_{ii}}{N}$, $a_{ii}$ represents the actual observation concordance number, and $N$ represents the total number of samples. $p_e = \sum_i p_{i\cdot} p_{\cdot i}$ is called the expected concordance rate, that is, the concordance rate of the two test results due to chance, where $p_{i\cdot} = \frac{R_i}{N}$, $p_{\cdot i} = \frac{C_i}{N}$, $R_i$, $C_i$ are the grand totals for rows and grand totals for columns of the i-th grid point respectively.

Table 3 comprehensively compares the three indicators of each origin identification model. It can be seen that the performance of decision trees, Naive Bayes model and RBF in the three indicators need to be improved compared to other models. LDA and PLS-DA perform well on the three indicators. Regardless of which evaluation indicator is used, the PCA-SVM combined model proposed in this paper performs the best among all models. Using this model to identify the origin of *Cornus officinalis*, the Accuracy is 84.8%.

## 5. Conclusions

The origin of Chinese Herbal Medicines is an important part of the quality control of Chinese Herbal Medicines, and it is also of great significance in the exploration and utilization of medicine sources [5, 13]. As a non-destructive analysis technique, mid-infrared spectroscopy has the advantages of short analysis time, simple operation, and low analysis cost. In recent years, it has received increasing attention in the identification of Chinese Herbal Medicines. In this study, a method for rapid origin identification of *Cornus officinalis* based on mid-infrared spectroscopy and chemometrics was established using the spectral data of *Cornus officinalis*. The research results showed that although the mid-infrared spectral information of the same *Cornus officinalis* has strong similarities, they also have certain differences in some parts. The spectral information is fully extracted by principal component analysis [53, 54], and the classification and identification model established by the support vector machine has a high accuracy. The predictive ability of the model was evaluated by an external test set, and the results showed that the established model could classify and identify 158 *Cornus officinalis* samples from 11 different regions with an accuracy rate of 84.8%. The accuracy of the external test set and validation set is similar, indicating that the model has strong generalization ability. Compared with the SVM model with full-spectrum data, the PCA-SVM model not only reduces the redundant variables in the model, but also has higher accuracy. In addition, by comparing with other commonly used stoichiometric models such as naive Bayes model, decision trees, LDA, RBF and PLS-DA, the PCA-SVM combined model performs the best among the three indicators given in this paper for the origin identification of *Cornus officials*. The method proposed in this paper can effectively shorten the identification time and cost of medicinal materials, and ensure the reliability of identification results. However, the scope of application of any model is limited by the sample space. Although the model established in this experiment shows good accuracy and robustness in both interactive and external tests, there is still much work to be done to promote it as a practical technique. Future studies can collect different

**Table 3. Comparison results of models for origin identification of *Cornus officinalis* based on mid-infrared spectroscopy.**

| Evaluation indicators | Naive Bayes | Decision Trees | LDA | RBF | PLS-DA | PCA-SVM |
|---|---|---|---|---|---|---|
| Accuracy (%) | 66.5 | 58.2 | 70.9 | 64.6 | 73.4 | 84.8 |
| F1-Score | 0.657 | 0.574 | 0.715 | 0.635 | 73.8 | 0.844 |
| Kappa | 0.628 | 0.538 | 0.678 | 0.609 | 70.3 | 0.831 |

classes of Chinese Herbal Medicines for research to improve the generalizability of the model. In addition, mid-infrared spectroscopy provides less information about the content of specific active constituents of the plant, and if more information on the content of specific active constituents of the plant is required, a more sophisticated analysis is required.

## Supporting information

**S1 File. Mid-infrared spectral dataset.**
(XLSX)

## Author Contributions

**Conceptualization:** Yueqiang Jin, Bing Liu.

**Formal analysis:** Yueqiang Jin, Bing Liu, Shasha Shi.

**Funding acquisition:** Chaoning Li.

**Investigation:** Bing Liu.

**Methodology:** Bing Liu.

**Project administration:** Bing Liu, Chaoning Li.

**Resources:** Yueqiang Jin, Bing Liu.

**Supervision:** Yueqiang Jin.

**Writing – original draft:** Yueqiang Jin.

**Writing – review & editing:** Yueqiang Jin, Bing Liu, Shasha Shi.

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
