## [Decision Letter · Decision Letter 0]

1 Dec 2022

PONE-D-22-30208Origin identification of Cornus officinalis based on PCA-SVM combined modelPLOS ONE

Dear Dr. Jin,

Thank you for submitting your manuscript to PLOS ONE. After careful consideration, we feel that it has merit but does not fully meet PLOS ONE’s publication criteria as it currently stands. Therefore, we invite you to submit a revised version of the manuscript that addresses the points raised during the review process.

We look forward to receiving your revised manuscript.

Kind regards,

Naji Arafat Mahat, PhD

Academic Editor

PLOS ONE

“This work was supported by the research project on philosophy and social science of universities in Jiangsu Province (No. 2022SJYB0562) and the horizontal scientific research project of Nanjing Vocational University of Industry Technology (No. HK22-38-01).”

5. Please upload a new copy of Figure as the Figure file cannot be open. Please follow the link for more information: https://blogs.plos.org/plos/2019/06/looking-good-tips-for-creating-your-plos-figures-graphics/" https://blogs.plos.org/plos/2019/06/looking-good-tips-for-creating-your-plos-figures-graphics/

Additional Editor Comments:

Please refer to the commented manuscript uploaded by Reviewer #1.

Reviewers' comments:

Reviewer's Responses to Questions

**Comments to the Author**

1. Is the manuscript technically sound, and do the data support the conclusions?

Reviewer #1: Partly

Reviewer #2: Partly

2. Has the statistical analysis been performed appropriately and rigorously? 

Reviewer #1: No

Reviewer #2: No

3. Have the authors made all data underlying the findings in their manuscript fully available?

Reviewer #1: No

Reviewer #2: No

4. Is the manuscript presented in an intelligible fashion and written in standard English?

Reviewer #1: Yes

Reviewer #2: Yes

5. Review Comments to the Author

Reviewer #1: The manuscript discusses the use of Infrared Spectroscopy (IR) combined with Principal Component Analysis (PCA) in tandem with Support Vector Machine (SVM) for origin identification of Chinese Herbal i.e. Cornus Officinalis collected from eleven different origin/provinces in China. I personally found the manuscript is interesting however major revision needs to be made to the manuscript prior to its publication in PLOS ONE.

My main concern pertaining the manuscript is that other classification models such as naive Bayes, Decision Tree, Linear Discriminant Analysis (LDA), Partial Least Square-Discriminant Analysis (PLS-DA) and a variant of artificial neural network i.e. Radial Basis Function (RBF) were reported but no works showing how these classification models were developed and tested in this study except Table 3 which briefly displays the precision and recall (sensitivity) of these classification models. Furthermore the reference i.e. reference 44 associated with these classification models was on rapid screening of chronic renal failure while reference 47 and 48 was study conducted to lamost dr6 and coffee samples respectively but not on the identification of Cornus Officinalis.

I was not able to locate all the figures mentioned in the manuscript, in other words figure were not included in the manuscript. This made the reviewing process quite difficult. Assuming that Figure 3 is available, the similarities and differences between the spectra of the Cornus Officinalis samples can be directly from the spectra therefore reporting correlation using Pearson Linear Correlation Coefficient in my opinion isn't necessary.

The explanation on data pre-processing was confusing. Authors mentioned that data pre-processing is the first step in modeling and also mentioned that the first step in constructing SVM based identification model is organize, collect sample followed by data normalization however since AU of the spectra did not deviate much from the Beer's Lambert Law, no special treatment was made to the data by the authors. Does this mean that authors used raw data for PCA? Since SVM was developed using principal components extracted from the spectral data, does this mean that authors normalized the principal components prior to SVM?

Authors should focus on reporting the outcomes acquired from PCA and SVM and disregard comparing with other classification models unless one of the objectives of the manuscript was to develop and compare different classification models for identification of Cornus Officinalis. Since authors mentioned the study compared different SVM kernel functions i.e. linear, quadratic, cubic and Gaussian, it would be interesting if the authors could report on these.

Other minor comments are attached in the manuscript.

Reviewer #2: 1. Authors presented the work that combines two multivariate techniques for source determination of a herbal medicine plant from numerous geographical origin.The work is interesting however the structure of the manuscript and important information are missing or if presented, it is not clearly described. some basic practice for writing scientific article was also not practised, thus improvements to the writing, contents and continuity of the article, should be made if the authors intent to publish the manuscript.

Simple corrections that authors can easily made are to italicised the genus and species of herbal plants where applicable, in text citations without initials, improper citation in text, capital letters and small letters (typo perhaps), figure mentioned but not included, make and model of the instrument and font sizes are among the typesetting that should be given attention by the authors. Those are some minor but vital components that would warrant a complete and well prepared manuscript.

2. In the introduction section, authors mention that microscopic analysis is able to determine authenticity of the herbs even when the herbs are highly processed. If this is the case then what is the need for multi varied analysis? Authors also did not mention specifically which part of the plant that was taken for analysis. It seems that the authors are generalising that all the part from that plants are the same. Is this true?

The data obtained seems to be the secondary data obtained from other organization and has been mentioned however the link given is confusing. The purpose of the link was not explained.

In term of statistical analysis, data collection is not clearly explained. Perhaps geographical origin can be better explain with sampling map so that the reader would have better understanding of the location where samples were collected as this is discussed in depth by the authors.

In data source and preprocessing, since this study aims to determine the origin of the sample, perhaps consideration of variations within and between (reproducibility and repeatability issues) of the samples should be considered. The authors merely state their opinions on variations observed to the signal but did not perform any testing or measurement to their hypothesis. The authors can be more specific when discussing the characteristic parts of the FTIR spectrum responsible for the source determination.

Authors have included many statistical techniques tested on Chinese Herbal medicine origin, but are the samples are of Cornus officials? It is true that FTIR spectra may show different spectral characteristic due to different chemical constituents within the sample however to comparison different samples of Chinese herbal medicine (for example quoted as reference 44) to Cornus officials is unfounded. It is also unclear whether the author perform naives Bayes, decision tree, LDA, RBF and PLS DA to the same data set used for PCA-SVM because it was not indicated in the methodology.

Based on this comments, it is highly suggested for the authors to revise the manuscript especially interm of explaining the work to avoid confusion, improve on the contents continuity and arrangement.

6. PLOS authors have the option to publish the peer review history of their article (what does this mean?). If published, this will include your full peer review and any attached files.

Reviewer #1: No

Reviewer #2: No

---

## [Author Response · Author response to Decision Letter 0]

12 Jan 2023

Dear Editor and Reviewers,

Thank you very much for your careful review and constructive suggestions with regard to our manuscript "Origin identification of Cornus officinalis based on PCA-SVM combined model" (Submission ID PONE-D-22-30208). Those comments are all valuable and very helpful for revising and improving our paper, as well as the important guiding significance to our researches. I have tried our best to revise the manuscript according to the reviewers’ comments, which are marked in red in this revised version. I appreciate for editor and reviewers’ warm work earnestly, and hope that the corrections will meet with approval. The main corrections in our manuscript and the responds to the reviewers' comments are presented as follows.

Reviewer 1

The manuscript discusses the use of Infrared Spectroscopy (IR) combined with Principal Component Analysis (PCA) in tandem with Support Vector Machine (SVM) for origin identification of Chinese Herbal i.e. Cornus officinalis collected from eleven different origin/provinces in China. I personally found the manuscript is interesting however major revision needs to be made to the manuscript prior to its publication in PLOS ONE.

1. My main concern pertaining the manuscript is that other classification models such as naive Bayes, Decision Tree, Linear Discriminant Analysis (LDA), Partial Least Square-Discriminant Analysis (PLS-DA) and a variant of artificial neural network i.e. Radial Basis Function (RBF) were reported but no works showing how these classification models were developed and tested in this study except Table 3 which briefly displays the precision and recall (sensitivity) of these classification models. Furthermore the reference i.e. reference 44 associated with these classification models was on rapid screening of chronic renal failure while reference 47 and 48 was study conducted to lamost dr6 and coffee samples respectively but not on the identification of Cornus officinalis.

Response: Thank you very much for your comment. As you mentioned, the main work of this study is to discuss the application of Infrared Spectroscopy (IR) with Principal Component Analysis (PCA) and Support Vector Machine (SVM) in the origin identification of Cornus officinalis. After completing the PCA-SVM model, this model was then compared with other commonly used origin identification models, and it was found that the PCA-SVM model performed better in both the training set and the test set.

For the other classification models, the manuscript is briefly described in Section 4. In the model development and testing, the sample divisions of the training and test sets for each model and the model validation are done in the same way as the combined PCA-SVM model. The same data and the same divisions are compared so that the superior performance of the PCA-SVM model can be demonstrated. The development and testing of other models are described and marked in red at the beginning of Section 4, paragraph 7 of our manuscript.

References [44], [47] and [48] did not identify the origin of Cornus officinalis. As you suggested, it would have been more convincing if the manuscript had introduced other models for the identification of Cornus officinalis origin using infrared spectroscopy when citing references. However, we encountered difficulties in accessing the references. There are very few literatures that both use infrared spectroscopy and identify the origin of Cornus officinalis and use these commonly used classification models. The references introduced into the manuscript did not identify the origin of Cornus officinalis, but used these common classification models associated with the manuscript, so we thought they could be introduced into the manuscript. In addition, precisely because there is little literature identifying the origin of Cornus officinalis, this reflects the significance of our manuscript publication.

2. I was not able to locate all the figures mentioned in the manuscript, in other words figure were not included in the manuscript. This made the reviewing process quite difficult. Assuming that Figure 3 is available, the similarities and differences between the spectra of the Cornus officinalis samples can be directly from the spectra therefore reporting correlation using Pearson Linear Correlation Coefficient in my opinion isn't necessary.

Response: Thank you very much for your suggestion. We checked the manuscript and it is true that there are no figures in the manuscript. This may have been an error in our submission of the manuscript. We apologize for the difficulties this has caused you in reviewing it. Figure 3 provides a visual representation of the differences between the sample spectra. Our purpose of using Pearson Linear Correlation Coefficient to report correlations is to quantify the differences between the sample spectra. As you said, this may make the manuscript look somewhat repetitive and not concise enough. We have removed the Pearson linear correlation coefficient section as you suggested, so that the manuscript looks more concise and clear.

3. The explanation on data pre-processing was confusing. Authors mentioned that data pre-processing is the first step in modeling and also mentioned that the first step in constructing SVM based identification model is organize, collect sample followed by data normalization however since AU of the spectra did not deviate much from the Beer's Lambert Law, no special treatment was made to the data by the authors. Does this mean that authors used raw data for PCA? Since SVM was developed using principal components extracted from the spectral data, does this mean that authors normalized the principal components prior to SVM?

Response: Thank you very much for your comment. We apologize for any confusion caused by the data pre-processing. In this paper, the data processing is carried out in the following steps. First, the collected data are first processed for outliers and missing values. We consider data that are less than 1/3 times the arithmetic mean of the nearest neighboring values on the left and right or greater than 3 times the arithmetic mean of the nearest neighboring values on the left and right as outliers. Outliers and missing values are interpolated by means of mean interpolation in this paper. 

After dealing with missing values and outliers, the absorbance data of 3448 corresponding bands under spectral illumination were analyzed and summarized. The range of absorbance after summary is -0.00675~1.48696 AU. 

Since the absorbance used for the data in this paper is an instrument-corrected value, some negative values appear. However, the absolute values of these negative values are small, and the total amount of negative values is less than 0.001% of the total amount of data, we keep them. The absorbance of some data is greater than 1 AU, but the maximum value of absorbance does not exceed 1.5 AU. We think that they do not deviate from the Lambert-Beer law, so no special treatment is required.

Our collection of mid-infrared spectral data contains 3448 bands, and the absorbance data for each band are highly correlated. If all 3448 variables are introduced into the Cornus officinalis origin identification model, it will not only make the model training time very long, but also the introduction of highly correlated variables into the model will lead to poor stability and generalization ability. Therefore, it is necessary to use mathematical methods to extract features from the data. Principal component analysis was used in this study to extract the features of the spectra. According to the Kaiser criterion, the first 14 principal components are selected to replace the original variables to establish the origin identification model of Cornus officinalis.

The 14 principal components after feature extraction were used to build the origin identification model of Cornus officinalis with the help of support vector machine. We name this model as PCA-SVM combined model. In the process of building the SVM classification model, as you mentioned, the extracted components need to be normalized. We have added a description of this in paragraph 4 of section 3.3 of the manuscript and marked it in red. 

4. Authors should focus on reporting the outcomes acquired from PCA and SVM and disregard comparing with other classification models unless one of the objectives of the manuscript was to develop and compare different classification models for identification of Cornus officinalis. Since authors mentioned the study compared different SVM kernel functions i.e. linear, quadratic, cubic and Gaussian, it would be interesting if the authors could report on these.

Response: Thank you very much for your comment. As you said, one of the objectives of this paper is to develop and compare different classification models to identify the origin of Cornus officinalis. Therefore, we have compared the results for other commonly used models. We strongly agree with your mention of reporting these different support vector machine kernel functions, which will make our manuscript more complete to read. We have added these common kernel functions in Section 3.3, paragraph 5 of the manuscript and marked them in red.

5. Other minor comments: In section 3.3, paragraph 2: No matter which set, the ratio of the maximum number and the minimum number of samples does not exceed 4:1, so it is considered that there is no sample imbalance in this study. 

Comment: Was this based on previous study?

Response: Thank you very much for your comment. Sample imbalance refers to the situation where the number of training samples of different classes in a classification task varies significantly. In general, an imbalance ratio (majority class vs. minority class) significantly greater than 1:1 (such as 10:1) can be classified as a problem of sample imbalance. The unbalanced sample will lead to the actual prediction with a focus on majority class, resulting in better accuracy in majority class and worse accuracy in minority class. Sample imbalance can generally be handled by resampling the dataset, generating artificial data samples, trying different classification algorithms, and penalizing the model

 Considering that the ratio of both the maximum and minimum sample size of Cornus officinalis did not exceed 4:1 and the number of origin categories was relatively high, based on our team's experience, we concluded that there was no sample imbalance in this study.

Reviewer 2

1. Authors presented the work that combines two multivariate techniques for source determination of a herbal medicine plant from numerous geographical origin.The work is interesting however the structure of the manuscript and important information are missing or if presented, it is not clearly described. some basic practice for writing scientific article was also not practised, thus improvements to the writing, contents and continuity of the article, should be made if the authors intent to publish the manuscript.

Simple corrections that authors can easily made are to italicised the genus and species of herbal plants where applicable, in text citations without initials, improper citation in text, capital letters and small letters (typo perhaps), figure mentioned but not included, make and model of the instrument and font sizes are among the typesetting that should be given attention by the authors. Those are some minor but vital components that would warrant a complete and well prepared manuscript.

Response: Thank you very much for your comment. We strongly agree with your suggestions for the structure of the manuscript to be continuous, the important information of the paper to be kept intact, and the writing of the paper to be standardized. A complete and standardized manuscript is more conducive to the publication and promotion of its contents. We have added important information to the manuscript as you suggested, and have corrected incorrectly quoted text and highlighted it in red.

2. In the introduction section, authors mention that microscopic analysis is able to determine authenticity of the herbs even when the herbs are highly processed. If this is the case then what is the need for multi varied analysis? Authors also did not mention specifically which part of the plant that was taken for analysis. It seems that the authors are generalising that all the part from that plants are the same. Is this true?

Response: Thank you very much for your comment. Microscopic identification is a term in Chinese medicine published in 2004. It is a method that uses microscope to observe the internal tissue structure, cells and the morphology of cellular contents of drugs, describe microscopic features, and develop microscopic identification basis to identify the authenticity of drugs. This identification method is fast, sensitive and simple, and has some practical significance. However, microscopic identification also has some shortcomings, such as the microscopic characteristics of some herbs are not easy to search, and the identification characteristics of some herbs do not conform to the pharmacopoeia. With the development of chromatographic coupling technology, the use of modern chromatographic techniques for the examination of Chinese herbal medicines has also been rapidly popularized. Infrared spectrometry has the advantages of fast analysis, low cost, non-destructive and simple pre-treatment. In recent years, it has been widely used in the field of quality control of Chinese herbal medicines. We have added a note in part 1, paragraph 2 of the manuscript and marked it in red.

As you mentioned, we did not mention in the manuscript the specific part of the plant from which the analysis was taken. The spectral data extracted in the manuscript is the dried and ripe pulp of Cornus officinalis, and we have added a note in the first paragraph of Section 2.1 of the manuscript and marked it in red. We apologize for any trouble caused by our oversight.

3. The data obtained seems to be the secondary data obtained from other organization and has been mentioned however the link given is confusing. The purpose of the link was not explained.

Response: Thank you very much for your comment. As you said, the data used in this study are secondary data obtained from other organizations. We have provided the relevant data in the relevant document, so giving the link loses its relevance. We have removed the link from the manuscript as you suggested.

4. In term of statistical analysis, data collection is not clearly explained. Perhaps geographical origin can be better explain with sampling map so that the reader would have better understanding of the location where samples were collected as this is discussed in depth by the authors.

Response: Thank you very much for your comment. According to your comment, we checked the manuscript, which states 11 origins, but does not clearly give the correspondence between the origins and OP1~OP11. In fact, they correspond in order, and we have added a note in section 1, paragraph 5 of the manuscript and marked it in red.

5. In data source and preprocessing, since this study aims to determine the origin of the sample, perhaps consideration of variations within and between (reproducibility and repeatability issues) of the samples should be considered. The authors merely state their opinions on variations observed to the signal but did not perform any testing or measurement to their hypothesis. The authors can be more specific when discussing the characteristic parts of the FTIR spectrum responsible for the source determination.

Response: Thank you very much for your suggestion. We strongly agree with you on the issue of test reproducibility and repeatability, if a test is not reproducible and repeatable, then the test is meaningless.

The reproducibility of the test is evaluated by taking 5 consecutive measurements of a given sample and calculating the Relative Standard Deviation (RSD) of their maximum common peak wave number. The RSD of the reproducibility test is determined to be less than 0.2%, indicating good reproducibility of the test. The repeatability of the test is evaluated by measuring the same sample once by 5 different experimenters, and calculating the RSD of the maximum common peak wave number of 5 measurements. The RSD of the repeatability test is determined to be less than 3%, indicating good repeatability of the test. We have added a note in Section 2.1, paragraph 2 of the manuscript and marked it in red.

6. Authors have included many statistical techniques tested on Chinese Herbal medicine origin, but are the samples are of Cornus officials? It is true that FTIR spectra may show different spectral characteristic due to different chemical constituents within the sample however to comparison different samples of Chinese herbal medicine (for example quoted as reference 44) to Cornus officials is unfounded. It is also unclear whether the author perform naives Bayes, decision tree, LDA, RBF and PLS-DA to the same data set used for PCA-SVM because it was not indicated in the methodology.

Response: Thank you very much for your comment. We have included in the manuscript many references to statistical techniques for the origin of herbs, many of which do not directly identify the origin of Cornus officinalis. As you suggested, it would have been more convincing if the manuscript had introduced other models for the identification of Cornus officinalis origin using infrared spectroscopy when citing references. However, there is very little literature using these statistical techniques and using infrared spectroscopy to directly identify the origin of Cornus officinalis. Although the references in the manuscript did not directly identify the origin of Cornus officinalis, they used these statistical techniques to identify the origin of other medicinal herbs. Therefore, in the context of the more difficult search for literature identifying the origin of Cornus officinalis, we introduced them into the manuscript for comparison with the combined PCA-SVM model proposed in this paper. The comparison results showed that the PCA-SVM combined model proposed in this paper performed well in each of the given evaluation metrics. In addition, precisely because there is little literature identifying the origin of Cornus officinalis, this reflects the significance of our manuscript publication.

For the development and testing of the naives Bayes, decision tree, LDA, RBF and PLS-DA models, the sample division of the training and test sets for each model and the validation of the models were performed in the same way as for the PCA-SVM combined model. The same data and the same divisions are compared so that the superior performance of the PCA-SVM model can be demonstrated. The development and testing of other models are described and marked in red at the beginning of Section 4, paragraph 7 of our manuscript.

Finally, thanks again to the editor for giving me the opportunity to revise the paper. Your serious and responsible attitude deserves our admiration. Thanks to reviewer 1 for the comments. Your encouragement provides the motivation for our team to move forward. Thanks to reviewer 2 for the suggestion. Your rigorous academic attitude has provided us with great help in future thesis writing.

Sincerely,

Best regards,

Yueqiang Jin

---

## [Decision Letter · Decision Letter 1]

15 Feb 2023

Origin identification of Cornus officinalis based on PCA-SVM combined model

PONE-D-22-30208R1

Dear Dr. Jin,

We’re pleased to inform you that your manuscript has been judged scientifically suitable for publication and will be formally accepted for publication once it meets all outstanding technical requirements.

Kind regards,

Naji Arafat Mahat, PhD

Academic Editor

PLOS ONE

Additional Editor Comments (optional):

Reviewers' comments:

Reviewer's Responses to Questions

**Comments to the Author**

1. If the authors have adequately addressed your comments raised in a previous round of review and you feel that this manuscript is now acceptable for publication, you may indicate that here to bypass the “Comments to the Author” section, enter your conflict of interest statement in the “Confidential to Editor” section, and submit your "Accept" recommendation.

Reviewer #1: All comments have been addressed

Reviewer #2: All comments have been addressed

2. Is the manuscript technically sound, and do the data support the conclusions?

Reviewer #1: Yes

Reviewer #2: Yes

3. Has the statistical analysis been performed appropriately and rigorously? 

Reviewer #1: Yes

Reviewer #2: Yes

4. Have the authors made all data underlying the findings in their manuscript fully available?

Reviewer #1: Yes

Reviewer #2: Yes

5. Is the manuscript presented in an intelligible fashion and written in standard English?

Reviewer #1: Yes

Reviewer #2: Yes

6. Review Comments to the Author

Reviewer #1: (No Response)

Reviewer #2: Corrections and explanation have been done by authors. I am satisfied with the manuscript and hope that the authors will keep on producing a good research and share the knowledge to the rest of the scientific community.

7. PLOS authors have the option to publish the peer review history of their article (what does this mean?). If published, this will include your full peer review and any attached files.

Reviewer #1: No

Reviewer #2: No

---

## [Editor Report · Acceptance letter]

17 Feb 2023

PONE-D-22-30208R1 

Origin identification of *Cornus officinalis* based on PCA-SVM combined model 

Dear Dr. Jin:

I'm pleased to inform you that your manuscript has been deemed suitable for publication in PLOS ONE. Congratulations! Your manuscript is now with our production department. 

Kind regards, 

on behalf of

Dr. Naji Arafat Mahat 

Academic Editor

PLOS ONE